# Corporate Social Responsibility, CEO Compensation Structure, and Corporate Innovation Activities

**Bu-Kyung Choi [1], Ji-Young Ahn [1,*] and Myeong-Cheol Choi [2,*]**

1   School of Business, Ewha Womans University, 52 Ewhayeodae-gil, Seoul 03760, Korea; cbk1903@naver.com
2   College of Business, Gachon University, Seongnam 13120, Korea
*   Correspondence: jy-ahn@ewha.ac.kr (J.-Y.A.); oz760921@gachon.ac.kr (M.-C.C.)

**Abstract:** This study empirically investigated the economic effect of CSR initiatives on innovation by examining Korean firms. Our primary objective of this study was to explore how a CEO compensation system can affect the CSR-innovation relationship. An integrated model of the impact of CSR on innovation activities was developed through analyzing various CEO compensation components such as structure, type, mix, and distribution. We identified the CEO compensation system that more suitably supports CSR in driving innovation performance improvement, and empirically examined a compensation system that enhances corporate innovation by creating a good alignment with CSR. Using a longitudinal data, we empirically tested the interactive effect of a CSR and compensation system of CEO in Korean publicly traded companies. Our empirical findings concerning the interaction between CSR strategies and CEO compensation schemes hold practical implications for establishing and implementing a suitable human resource system to improve organizational competitiveness.

**Keywords:** corporate social responsibility; innovation; CEO compensation structure; social exchange; equity

## 1. Introduction

Corporate social responsibility (CSR) is defined as the obligation of a company to pursue and make decisions for policies that are desirable from the point of view of social goals and values [1]. In recent years, CSR has moved away from the narrowly defined imperatives of "social contribution" and "transparent management" toward fulfilling economic, environmental, and ethical responsibilities to support long-term corporate stability and sustainability in a dynamic environment. The definition is being expanded to "creating shared value," a strategic concept that stimulates sustainable growth [2]. Recent theoretical studies and anecdotal evidence suggest that CSR activities can lead to process and product innovation by promoting investment in research and development (R&D), a major driver of corporate innovation [3,4]. For example, some firms have developed new business models with special emphasis on product and service innovations through CSR activities [5], which can in turn lead to improve corporate image among consumers.

Despite the popularity of CSR activities as strategic initiatives to innovate and secure competitive advantage, empirical studies on CSR-innovation relationship are relatively absent or are characterized, at best, by mixed results [3,4]. Moreover, the CSR-innovation link has not received much attention in the Korean context. However, anecdotal evidence suggests that there is a growing number of listed companies in Korea that utilize CSR as a driving force for innovation, thereby enhancing the company's competitive advantage [5]. Thus, one purpose of the study was to empirically investigate the economy-wide average effect of CSR initiatives on innovation by examining all publicly traded firms in Korea. Furthermore, our primary objective of this study was to explore how CEO compensation system can affect the CSR-innovation relationship. The empirical inconsistency of the impact of CSR on innovation may indicate that firms differ widely in their ability to deal with the impact of CSR on innovative efforts. Additionally, it is expected that CEO

incentives play key role in influencing both CSR and corporate innovation decisions that are often associated with high-risk projects and continuously allocated resources for the sake of conflicts of interest among stakeholders [6]. Combined together, we argue that a certain choice of CEO compensation component may be better able to create a positive relationship between CSR and innovative activities.

This study explored a comprehensive understanding of the role of CEO compensation design in the relationship between CSR and innovation and presented empirical evidence to show the relationship. Specifically, certain elements of CEO compensation can increase the effect of CSR on innovation activities. As investment invariably accompanies CSR and corporate innovation, conflicts between management and shareholders regarding these activities frequently arise [6]. Therefore, CEO compensation must be designed—by considering elements such as risk premiums and incentives—keeping these possible conflicts of interest in mind to ensure that innovation performance from CSR is enhanced [7,8]. Several recent studies confirm that CEO salary level has a negative effect on CSR [9,10]. However, CEO compensation level alone is insufficient to explain the link between CSR and innovation activities. Therefore, in this study, an integrated model of the impact of CSR on innovation activities was developed through analyzing major CEO compensation design variables such as structure, type, mix, and distribution. Put differently, we theoretically explored the design variables of CEO compensation that more suitably support CSR in driving innovation performance improvement, and identified a compensation system that enhances corporate innovation by creating greater synergy and alignment with CSR.

Using a longitudinal data, we empirically tested the interactive effect of CSR and compensation system of CEO in Korean publicly traded companies, including compensation structure (percentage of performance-based compensation to total compensation), compensation type (the ratio of stock-linked performance to total pay for performance), and reward distribution (the wage gaps within the top management team and between the CEO and the average employee). We analyzed the compensation structure and details of individual, registered executives who received a remuneration of 500 million Korean won or more. Compensation details of executives meeting this criterion must be disclosed in accordance with the revision of the Capital Market Act at the end of 2013. Overall, this study proposed the theoretical and practical implications of managerial compensation structure for the impact of CSR on innovation.

## 2. Theoretical Background and Hypotheses

### 2.1. CSR and Innovation Activities

Innovation is the new adoption of a design, system, policy, program, process, product, or service created and purchased within an organization [11]. Recent studies have discussed that CSR has a positive effect on corporate innovation activities [12,13]. For example, a study by the European Commission found that CSR creates opportunities for companies to adopt a long-term, strategic approach to developing innovative products, services, and business models that provide higher quality and more productive jobs [14]. Importantly, CSR is reported to play a role in attracting individuals to explore [14,15]. In addition, CSR-driven innovation values oriented toward products and services with a social, environmental, or sustainability purpose. Specifically, companies often apply CSR to productive processes and practices that may involve R&D spending or require technological change [16]. For instance, Bansal [17] showed that innovations in processes and products have been pursued in order to reduce carbon dioxide emissions. As such, CSR provides opportunities for innovation and can be spurred by social, environmental, and sustainability drivers to create new ways of working, products, services, processes, and new market spaces.

Moreover, CSR can create a workplace environment in which external stakeholders actively participate, identify new business opportunities through active alternatives to social tasks, and encourage employees to think and work in more innovative ways [18]. Essentially, CSR activities help companies form new relationships and strengthen exist-

ing ones, thus, gaining broader access to important external information—such as the knowledge and expertise of various external stakeholders—that drives innovation. Additionally, external information complements internal knowledge and advances technological innovation capabilities [19,20].

Several studies have suggested a positive effect of CSR on R&D intensity [21,22]. Given that companies are more likely to address social, environmental, ethical, human rights, and consumer concerns through CSR activities, they enjoy trust and loyalty among stakeholders, including customers, employees, investors, business partners, and communities [2,23]. A positive relationship between CSR and innovation activities has been established. Therefore, based on findings from previous studies, we retest the following hypothesis:

**Hypothesis 1**. *CSR has a positive relationship with corporate innovation activities.*

### 2.2. CSR, CEO Compensation Structure, and Innovation Activities

#### 2.2.1. Performance Pay as a Percentage of Total CEO Compensation

The CEO is the chief decision-maker who not only manages the company on behalf of shareholders, but also takes responsibility for its performance [24]. CEO compensation is a representative governance mechanism that induces the company to move down and should be designed to be linked with the company's core activities and performance [25]. Total compensation for CEOs consists of base salary, cash bonus, and equity-based compensation [25,26]. The cash bonus is fixed-cash equivalent compensation—guaranteed by cash incentives and with a cap on compensation—meant as an additional reward for managers who reach a performance standard set by the company. In addition, stock-linked compensation is a generic term for all forms of reward in which employee compensation is linked to stock price, and typically includes stock options and grants [27]. This performance-based pay system refers to a method in which wages are differentially paid according to work performance, and is a way of supplementing the basic wage through the differential payment of bonuses and differential wage increases. Performance-based pay also occurs at the group level, when a performance bonus is paid uniformly to group members when target performance is achieved [9,10].

Most previous studies have examined the effect of CSR on the level of CEO compensation. For example, Cai et al. [28] investigated the impact of CSR on CEO compensation using US corporate data from 1996 to 2010. Their results stated that the CEO of a company with considerable CSR activity should receive less than the CEO of a company with a low level of CSR activity. Similarly, a later study demonstrated the differential effect of CSR based on the CEO's compensation type; base pay and long-term performance pay had a negative effect, but bonuses had no effect [29]. This was an analysis of the effects of separate individual compensation components; however, the compensation structure (i.e., performance pay as a proportion of total compensation) was not considered. Importantly, the reward structure is a direct variable in predicting the risk-seeking behavior of management. As both CSR and innovation activities are risk-bearing long-term investments, management will pursue less risky and short-term alternative strategies if compensation for increased risk is not offered [9].

Thus, CSR activities combined with incentivized risk-taking and behavior can generate synergistic effects on innovation activities. This is because the intensity of pay-for-performance pay can benefit from the alignment of organizational strategic goals and encouraging risk-taking behavior, thereby increasing the innovative climate. Overall, a high proportion of performance-based pay to total compensation is more effective in firms with CSR initiatives and such synergistic interaction has a positive impact on innovation activities. Thus, we propose the following hypothesis:

**Hypothesis 2.** *The positive relationship between CSR and innovation activities is stronger when the proportion of pay-for-performance pay to total compensation is high.*

### 2.2.2. Share of Equity-Based Pay in Total CEO Compensation

The types of performance pay discussed above do not always have similar interactions with CSR. CEO performance pay is divided into short- and long-term pay in terms of time frame [26,30]. Jensen and Murphy [26] showed that stock-related compensation is an effective compensation tool that incentivizes decision-making that improves the long-term performance of a company, given that the performance-based portion of the compensation is based on stock price. The cash bonus for CEOs in domestic companies is based on performance evaluation, while stock options are a typical example of a stock-based bonus.

It is expected that the CEO's preference for less risky short-term rewards is negatively related to CSR activities. For example, when CEOs invest in CSR for their own benefit and at the expense of shareholders, CSR performance is positively correlated with cash-based reward ratios [29]. In contrast, if the CEO engages in CSR to increase firm value in line with shareholder interests—or if successful CSR performance improves relationships with other stakeholders, including employees and shareholders—the share-based reward ratio increases according to social performance [29].

In addition, Gan and Park [31] argued for a positive effect between CEOs' management ability and stock-based compensation, supporting that stock option compensation for managers drives long-term decision-making by aligning the interests of shareholders and managers. Stock-based compensation reduces managers' compensation risk, thereby encouraging higher-risk investment decisions [32]. In other words, the more stock managers own, the less incentive compensation must be paid by shareholders [33]. Effective CSR activities are consistently conducted only when performance rewards are received [34–36].

Therefore, a compensation system with a higher proportion of stock-based long-term incentives—rather than with short-term incentives centered on cash bonuses—creates greater synergy with CSR. In particular, stock options, unlike stocks, are long-term incentives that must be held for at least three years after acquisition due to vesting period restrictions. Thus, as the proportion of stock-based incentives excluding cash incentives increases, CEOs adopt longer-term strategies consistent with activities such as CSR and investment in product innovation. Therefore, the relative weight of stock-based compensation to performance-based pay increases the effectiveness of CSR and has a positive effect on innovation performance. Thus, the following hypothesis is proposed:

**Hypothesis 3.** *The positive relationship between CSR and innovation activities is stronger when the proportion of stock-based compensation to total compensation is high.*

### 2.2.3. Pay Dispersion between CEO and Top Management Team

Pay dispersion refers to the degree of pay differential constituted by the pay structure of a company and is defined as the difference in the wages of workers within a job or organizational hierarchy [37,38]. In addition, the wage gap can be divided into a vertical wage gap—the wage difference between levels of an organization—and a horizontal wage gap—the wage difference between the same or similar jobs within the organizational hierarchy [37]. Executives within the top management team (TMT) work closely with the CEO to help formulate and implement company strategy, but are sometimes the fiercest contenders for succession to CEO [39–41]. TMT members are reasonably similar in their work experience, perspective, and temperament [42]. These executives are highly motivated and achievement-oriented, and tend to be influenced by the desire for power and position. This demonstrates a tendency to recognize the difference [39].

This study focused on the interaction between the pay differentials or inequity within TMTs and CSR. Although not specifically investigated in previous studies, it is believed that team incentives can reinforce specific attitudes and behaviors to realize the TMT's strategic goals, given that organizational leadership is a shared activity [43]. According to equity theory, individuals rethink the inputs (e.g., time, effort) to their work and the outputs (e.g., rewards) they receive from that work, and consider other tasks and inputs similar to themselves. In comparing ratios, social comparisons are made based on observable

differences, even among executives within the TMT [39,44]. This comparison demonstrates that when TMT members have strong similarities in performance, employment success, and hierarchical position within the firm [42], management overestimates the value of their contributions despite differences such as the quality and quantity of their work, and tends to underestimate the value of the efforts of others [39]. High wage disparities among TMT executives are associated with emotions such as negativity, apathy, social distancing, and jealousy, and with decreased job satisfaction and a high turnover rate [39,45].

Several previous studies have suggested that high wage gaps can lead to high executive turnover [46], which is particularly deleterious to performance in high-tech industries [45]. In addition, in a study of TMTs, the hierarchical reward gap hindered collaboration within peer groups as the reward gap increased, which can in turn lead to decrease organizational performance [45]. While Beaumont and Harris [47] demonstrated differences by industry and corporate ownership, their study supported that a narrow wage structure generally improved corporate performance. In addition, a study of long-term organizational performance supported that the compensation gap between the CEO and top executives had a negative effect on corporate innovation in 75 companies from 14 industries in the United States from 1991 to 1995 [48].

Overall, horizontal pay inequity can exacerbate competition among executives and undermine cooperative efforts, hindering the cooperation essential for CSR and innovation activities [49,50]. Thus, it is predicted that less horizontal pay inequity can provide more potential benefits to firms which pursue CSR by creating a cooperative culture within TMTs. Likewise, a high horizontal pay differential attenuates the positive interaction between CSR and innovation activities. Therefore, the following hypothesis is formulated:

**Hypothesis 4.** *The positive relationship between CSR and innovation activities is weaker when there is a high pay differential within the TMT.*

### 2.2.4. CEO-Employee Pay Dispersion

To date, CSR activities have been largely limited to endeavors targeting an organization's external stakeholders, such as shareholders, customers, business partners, and local communities [51]. However, given the recent interest in CSR activities for internal stakeholders (i.e., employees)—mainly in Korea and the United States—related research has been divided by internal versus external stakeholder perspectives. Specifically, internal CSR focuses on CSR activities within the organization to improve the lives and productivity of employees [52]. This process depends on establishing positive exchange relationships between employees and the organization [51–54]. However, the content and effectiveness of internal and external CSR activities may not always be consistent within an organization [55]. For example, in the case of companies that are very active in external CSR, these activities give a positive signal to external parties such as shareholders and customers; however, in some situations, they cause considerable cynicism among employees who are internal stakeholders, which in turn can weaken the effectiveness of CSR [56].

Employees frequently perceive fairness through wage comparisons, using CEO and TMT compensations as a reference [57,58]. Wage comparison among members of an organization influences identification and motivation through social identity by evaluating the adequacy of social exchange relationships [58,59]. According to social exchange theory, action of employee is related to the reaction of the management through reciprocity [57]. When a CEO in the organization fails to maintain a balance by not fulfilling their obligations, a breach of psychological contract occurs, which can result in negative attitudes such as job dissatisfaction, turnover intention, and negative behaviors [59]. The manager-employee pay multiple refers to the difference in the level of compensation between each organizational level. This implies that compensation is efficiently distributed based on managers' and employees' contribution to value creation in the company. When the imbalance in wage distribution creates a perception of inequality, a trust gap occurs between the CEO and employees [60]. This gap reduces employee commitment, cooperation, and

information-sharing fostered by the CEO, resulting in a negative impact on organizational performance [61]. In addition, an excessive wage gap between management and employees reduces mutual trust, as members of the same organization perceive that limited resources are exploitatively distributed [61,62]. The perception of unfair wage distribution decreases the attachment to the organization of the internal stakeholders of CSR and weakens trust, cohesion, and cooperation among members [63–66]. Thus, it is expected that less average wage gap between CEO and employees may provide more potential benefits to firms which pursue CSR by strengthening trust and cohesion within organizations. Put differently, a high average wage gap between CEO and employees weakens the positive interaction between CSR and innovation activities. Therefore, we formulated the following hypothesis:

**Hypothesis 5.** *The positive relationship between CSR and innovation activities is weaker when there is a high wage gap between CEO and employees.*

### 3. Research Method

*3.1. Sample Selection and Data Collection*

For an empirical analysis of the relationship between CSR and innovation, we selected companies continuously listed on the Korea Stock Exchange for five years from 2014 to 2018. Selection was limited to corporations with December settlement of accounts. To ensure sample homogeneity, we excluded the financial industry because of the difficulty in calculating R&D costs given the difference in the industry's use of an account classification system. Of note, the Financial Investment Services and Capital Markets Act in Korea was amended, and the study's sample period is from fiscal 2014 to fiscal 2018, where information on compensation of managers and individual executives of listed companies can be obtained [67]. From 2013, compensation details of executives earning 500 million won or more, in addition to comprehensive calculation criteria and methods, must be disclosed, by presidential decree, in accordance with the revision of the Capital Market Act [67].

The research sample for this study included companies for which data on CEO compensation, Environmental, Social and Governance (ESG) evaluation data, and other financial information were collected. Accordingly, the final sample was composed of panel data of 2936 CEO-year observations, after samples with missing information were excluded from the initial 3590 cases (five years of data for 718 companies).

In this study, the annual ESG evaluation score determined by the Korea Corporate Governance Service (KCGS) for listed domestic companies was used as a proxy for CSR activity level. The KCGS's ESG evaluation score comprehensively measures the level of CSR activity for each company [34]. In this study, the evaluation scores for the social responsibility (S) and environmental (E) management evaluation sections were used as proxy variables for CSR activities. Thus, the overall score consisted of five sub-evaluation areas. The score for each section had a maximum of 300 points.

Information on CEO compensation was sourced from the business report provided by the Financial Supervisory Service's Data Analysis of the Retrieval and Transfer (DART) system and the TS 2000 database of the Korea Listed Companies Association. Data on CEO compensation (total, cash, and stock) were gathered for companies obliged to disclose compensation details for individual registered executives in accordance with revisions to the Capital Market Act at the end of 2013. Data on employee wages and compensation and holdings of TMT members were collected from business reports provided by the Financial Supervisory Service's DART system.

Corporate innovation activities were identified from company business reports, the FnGuide, and the KIS-Value database. Lastly, information on patents was collected through the electronic patent acquisition system of the Financial Supervisory Service and the Korea Intellectual Property Rights Information Service, a comprehensive patent information service with a database of industrial property rights.

### 3.2. Variable Description and Measurement

The main variables and research models used in the empirical analysis are described in this section. A summary of the main variables is provided in Table 1.

**Table 1.** Definition of main variables.

| Variable Name | Variable Definition |
| --- | --- |
| CSR | The evaluation score of the Korea Corporate Governance Service, where E and S represent scores in the environmental and social responsibility management sectors, respectively, ES is the sum of the scores in the environmental and social responsibility management sectors. Sub-domains of the social responsibility evaluation section include S1, S2, S3, and S4, the evaluation scores for workers, suppliers and competitors, consumers, and communities, respectively. |
| R&D expenditure | Ratio of R&D expenditure to total sales [t + 1] |
| number of patents | Number of patents in the company [t + 1] |
| CEO-incentive | The natural logarithm of performance pay/CEO total compensation |
| CEO-Stock Performance | The natural logarithm of the ratio of stock performance pay to CEO pay-for-performance compensation |
| CEO-TMT wage gap | The natural logarithm of CEO total reward/TMT average reward |
| CEO-Employee wage gap | The natural logarithm of CEO total compensation/employee average compensation |
| CEO tenure | Year of CEO tenure (t) |
| business year | Years of company operation (t) |
| debt ratio | The company's debt ratio in the year (total liabilities/total assets) (t) |
| foreign | Foreign ownership rate |
| size | Firm size: the natural logarithm of the number of employees |
| Owner-manager | Owner-manager status = 1 if the CEO is the largest shareholder; 0 otherwise |
| CEO share | The natural logarithm of CEO stake |
| year/industry dummy | Industrial dummy changes according to the Korean standard industrial middle classification |

### 3.3. Research Model

In this study, we conducted panel data analyses. Panel data models provide information on individual firm (or CEO) behavior, both across individual firms and over time. The data and models have both cross-sectional and time-series dimensions. The panel data we used were unbalanced when firms were not observed in all time periods. As endogeneity may arise when estimating such a panel regression model, the Hausman test was performed to detect endogeneity between these explanatory variables and individual effects. Hausman test results can identify whether the models are random-effect or fixed-effect models. If the null hypothesis cannot be rejected, there is no explanatory variable or endogeneity problem. If the null hypothesis is rejected, it should be estimated using a fixed-effect model. In addition, we conducted normality tests such as Shapiro-Wilk normality tests, indicating normal distribution of data. For instance, the following is the empirical specification model 2 of Table 2 used in this study.

**Table 2.** Descriptive statistics and correlation analysis.

| | Mean | S.D | 1 | 2 | 3 | 4 | 5 | 6 | 7 | 8 | 9 | 10 | 11 | 12 | 13 |
|---|---|---|---|---|---|---|---|---|---|---|---|---|---|---|---|
| 1. R&D expenditure | 0.11 | 0.04 | | | | | | | | | | | | | |
| 2. Patent right | 18.9 | 135.3 | 0.24 *** | | | | | | | | | | | | |
| 3. CSR score | 5.44 | 1.07 | 0.15 *** | 0.36 *** | | | | | | | | | | | |
| 4. CEO performance pay ratio | 0.55 | 0.47 | 0.07 *** | 0.18 *** | 0.26 *** | | | | | | | | | | |
| 5. CEO share payout ratio | 0.45 | 0.46 | 0.10 *** | 0.08 *** | 0.14 *** | 0.11 *** | | | | | | | | | |
| 6. CEO-TMT wage gap | 3.94 | 1.94 | 0.06 ** | 0.01 | −0.01 | 0.05 ** | 0.05 ** | | | | | | | | |
| 7. CEO-Employee wage gap | 43.4 | 9.97 | 0.10 | 0.09 *** | 0.12 ** | 0.31 *** | 0.02 | 0.16 *** | | | | | | | |
| 8. Number of years of service | 15.64 | 14.06 | −0.010 | −0.05 ** | −0.18 *** | 0.01 | −0.02 | 0.03 * | −0.01 | | | | | | |
| 9. Company size | 1620.3 | 5803.5 | 0.05 ** | −0.08 *** | −0.04 * | 0.09 *** | 0.06 ** | 0.02 | 0.03 * | 0.22 *** | | | | | |
| 10. Corporate year | 39.79 | 19.43 | −0.03 | −0.03 * | −0.03 | −0.03 * | −0.05 ** | −0.02 | −0.05 ** | 0.07 *** | 0.05 ** | | | | |
| 11. Debt ratio | 122.55 | 404.09 | 0.02 | 0.04 * | 0.04 * | −0.01 | −0.05 ** | 0.01 | 0.01 | −0.07 ** | 0.03 | −0.05 ** | | | |
| 12. Foreign Investment ratio | 10.36 | 13.25 | 0.11 *** | 0.26 *** | 0.37 *** | 0.27 *** | 0.18 *** | 0.03 * | 0.12 *** | −0.02 * | −0.04 ** | −0.05 ** | −0.07 ** | | |
| 13. owner-manager | 0.47 | 0.50 | −0.09 *** | 0.006 | −0.11 *** | 0.07 *** | −0.002 | 0.003 | 0.05 * | 0.40 *** | 0.05 ** | 0.09 *** | −0.03 | −0.03 | |
| 14. CEO stake | 0.7 | 11.2 | −0.05 ** | −0.03 | −0.02 | 0.03 | 0.03 | −0.003 | −0.08 ** | 0.06 ** | −0.004 | 0.01 | −0.01 | −0.02 | 0.05 ** |

* $p < 0.1$; ** $p < 0.05$; *** $p < 0.01$. Note: * CSR score: converted to ESG evaluation score (A+: 10, A: 9, A−: 8, B+: 7, B: 6, B: 5); * R&D expenditure (KRW): R&D expenditure total expenditure/revenue; * CEO incentive pay (KRW): CEO performance bonus/CEO total compensation; * CEO stock performance bonus: CEO stock performance bonus/CEO total compensation; * CEO-TMT wage gap: CEO total compensation/TMT average compensation; * CEO-employee wage gap: CEO total compensation/employee average compensation; * Company size: number of employees; * CEO ownership ratio: stake owned by management; * Owner presence: dummy variable of 1 if owner; 0 otherwise.

$$ln[R\&D\ it + 1] = \beta_0 + \beta_1\ CSR\ it + \beta_2\ CEO\ performance\ pay\ ratio\ it + \beta_3\ (CSR\ *\ CEO\ performance\ pay\ ratio)\ it + \beta_4\ (Controls)\ it + \varepsilon it \tag{1}$$

## 4. Results

Table 2 presents the descriptive statistics and correlations for variables in the study. Tables 3 and 4 present the results of the panel regression model; Table 3 shows the results of effect of the explanatory variables on R&D expenditure. In Model 1, the CSR score had a significant positive effect ($\beta = 0.26$, $p < 0.01$) on the level of R&D expenditure. The coefficient of CSR on patent rights in Model 1 of Table 4 was also significantly positive. Therefore, Hypothesis 1 is supported, indicating that the higher the level of CSR activities of a company, the higher the innovation activities.

Hypotheses 2, 3, 4 and 5 predict the interaction effects of CSR and different components of CEO compensation system on innovative activities. First, we investigated the interaction between CSR and the proportion of performance-based pay to total compensation. As shown in Model 2 of Table 3, the interaction effect of CSR score and CEO performance pay was statistically significant and positive ($\beta = 0.72$, $p < 0.01$). Likewise, when patent rights were used as dependent variables, the coefficient value of interaction term was positive, as shown in Model 2 of Table 4. The results indicate that CSR combined with the high proportion of performance-based pay to total compensation can increase innovation activities, which supports Hypothesis 2.

Moreover, in Model 3, the coefficient value of the interaction term between CSR score and the proportion of CEO stock-based pay to total performance-based pay was significantly positive ($\beta = 1.77$, $p < 0.01$). Although the main effect of CEO stock-based pay was negative, the predicted interactive effect of CSR on innovation was still positive by offsetting the main effect, indicating that there is a positive effect of CSR and the intensity of CEO stock-based incentive on R&D expenditure. However, the interactive effect of CSR and the proportion of stock-based pay to total performance-based compensation on patent rights presented in Model 3 of Table 4 is not statistically significant. Thus, there is limited evidence supporting the alignment between CSR and the intensity of stock-based pay predicted by Hypothesis 3.

**Table 3.** CSR, innovation activities (R&D expenditure) and CEO compensation.

|  | Model 1 | Model 2 | Model 3 | Model 4 | Model 5 |
|---|---|---|---|---|---|
| Tenure | 0.47 * | 0.49 * | 0.47 * | 0.91 ** | 0.45 ** |
| Corporate year | 0.87 ** | 0.95 ** | 0.91 ** | 0.04 | 0.83 |
| Debt ratio | −0.001 | −0.001 | −0.002 | −0.001 | −0.001 |
| Foreign Investment Ratio | 0.09 *** | 0.06 ** | 0.06 ** | 0.09 ** | 0.07 ** |
| Company size | 0.31 ** | 0.25 ** | 0.30 ** | 0.24 ** | 0.32 *** |
| Owner-manager | −2.75 *** | −2.72 *** | −2.81 *** | −2.32 ** | −2.74 *** |
| CEO share | −0.33 ** | −0.236 ** | −0.37 ** | 0.01 | −0.34 ** |
| Year fixed effect | Yes | Yes | Yes | Yes | Yes |
| Industry fixed effect | Yes | Yes | Yes | Yes | Yes |
| CSR score | 0.26 *** | 0.29 * | 0.31 * | 0.13 *** | 0.27 ** |
| Performance pay |  | −0.12 ** |  |  |  |
| CSR * performance pay |  | 0.72 ** |  |  |  |
| CEO stock-based pay ratio |  |  | −1.17 ** |  |  |
| CSR * stock-based pay ratio |  |  | 1.77 ** |  |  |
| CEO-TMT wage gap |  |  |  | 0.04 |  |
| CSR * TMT wage gap |  |  |  | −0.002 |  |
| CEO-employee wage gap |  |  |  |  | −0.23 ** |
| CSR * CEO-employee wage gap |  |  |  |  | −1.13 ** |
| Adjusted $R^2$ | 0.05 | 0.06 | 0.05 | 0.04 | 0.09 |
| F value | 11.25 | 10.35 | 10.25 | 10.60 | 12.39 |
| Number of observations | 1732 | 1731 | 1732 | 1732 | 1733 |

* $p < 0.1$; ** $p < 0.05$; *** $p < 0.01$.

**Table 4.** CSR, innovation activities (patent rights) and CEO compensation.

|  | Model 1 | Model 2 | Model 3 | Model 4 | Model 5 |
|---|---|---|---|---|---|
| Tenure | 0.01 | 0.02 | 0.02 | −0.07 | −0.01 |
| Corporate year | −0.05 | −0.04 | −0.05 | 0.12 * | −0.03 |
| Debt ratio | 0.001 * | 0.002 * | 0.002 * | 0.002 | 0.002 ** |
| Foreign Investment Ratio | 0.02 *** | 0.01 *** | 0.02 *** | 0.03 *** | 0.03 *** |
| Company size | 0.02 ** | 0.03 ** | 0.02 | 0.03 ** | 0.03 *** |
| Owner-manager | 0.05 | 0.06 | 0.06 | −0.05 | 0.09 |
| CEO share | −0.06 *** | −0.06 *** | −0.05 *** | −0.002 | −0.003 |
| Year | Yes | Yes | Yes | Yes | Yes |
| Industrial dummy | Yes | Yes | Yes | Yes | Yes |
| CSR score | 0.21 *** | 0.16 *** | 0.24 *** | 0.14 *** | 0.17 *** |
| Performance-based pay ratio |  | −0.14 *** |  |  |  |
| CSR * performance pay ratio |  | 0.09 *** |  |  |  |
| CEO stock performance pay ratio |  |  | 0.15 |  |  |
| CSR * stock performance |  |  | 0.09 |  |  |
| CEO-TMT wage gap |  |  |  | −0.50 ** |  |
| CSR * TMT wage gap |  |  |  | −0.32 ** |  |
| CEO-employee wage gap |  |  |  |  | −0.33 ** |
| CSR * CEO-employee wage gap |  |  |  |  | −0.20 ** |
| Adjusted $R^2$ | 0.13 | 0.14 | 0.13 | 0.21 | 0.14 |
| F value | 30.02 | 25.17 | 24.17 | 17.14 | 25.62 |
| Number of observations | 1489 | 1488 | 1489 | 602 | 1489 |

* $p < 0.1$; ** $p < 0.05$; *** $p < 0.01$.

Third, Hypothesis 4 proposes the interactive relationship between CSR and TMT pay differentials. As shown in Model 4 of Table 3, the regression coefficient was not statistically significant. However, the coefficient value of interaction term between CSR and pay differential on patent acquisition presented in Model 4 of Table 4 was significantly negative ($\beta = -0.32$, $p < 0.01$). The results indicate limited evidence supporting the positive synergistic effect between CSR and TMT horizontal pay equity.

Finally, the potential interactive relationship between CSR and CEO-employee wage gap was strongly supported as predicted by Hypothesis 5, suggesting a good match between CSR and low CEO-employee wage gap. The regression coefficient of the interaction term of CSR and the CEO-employee wage gap was negative ($\beta = -1.13$, $p < 0.01$), as shown in model 5 of Table 4. It also confirms the interaction effect of CSR and wage gap on patent rights.

## 5. Discussion and Conclusions

Our first purpose in this study was to examine the impact of CSR initiatives on innovation activities. Hypothesis 1 predicted the positive effect of CSR on innovation. Based on the panel data of 718 publicly traded Korean firms and on archival financial and patent data for five years, we found strong evidence of the significant impact of CSR on innovation activities measured by R&D expenditure and patent rights acquisition. The results suggested that the more that companies engage in CSR activities, the more they conduct diverse social contribution activities utilizing the company's core competencies [68]. This is consistent with recent findings by Zhou et al. [69], suggesting CSR had a positive effect on service and product innovation of manufacturing companies in China. Similarly, a recent study conducted using multi-national sample suggested that the impact of CSR on corporate innovation is more pronounced in developed countries including Korea [70]. Likewise, these CSR-oriented corporate innovation activities fulfill environmental and social goals by offering innovative products and services and building corporate value. Thus, CSR activities can be the most important drivers of corporate innovation and play an important role in stimulating new product innovation [4].

Our primary objective was to understand how CEO compensation system affects CSR's impact on corporate innovation activities. First, findings indicated that the greater the proportion of performance-based pay to total compensation, the stronger the positive relationships between CSR and innovation activities, which supports Hypothesis 2. We may predict an increase in R&D investments of a company with a rise in the CEO's performance-based compensation. In other words, more compensation is provided to managers to incentivize investment in R&D. CSR and innovation activities may be regarded as decision-makers who take strategic decisions on behalf of shareholders [71].

Furthermore, we predicted that stock-based portion in the CEO performance-based compensation mix can strengthen the positive relationship between CSR and innovation activities. However, there is limited evidence supporting the potential alignment between CSR and the intensity of stock-based pay predicted by Hypothesis 3. The results may imply that compensation schemes based on stock price performance are long-term rewards that may motivate managers to look beyond the short-term perspective and make long-term decisions [72–74], which can align with the value of CSR, thereby leading to innovation activities. The results confirm this argument when patent was used as a dependent variable. This is partially consistent with the prior study, arguing that long-term portion of CEO compensation had a positive and significant effect on corporate CSR participation [75]. However, we do not find strong evidence supporting this relationship.

Additionally, our final goal was to explore how the distribution of CEO compensation affects the CSR-innovation relationship. Hypothesis 4 predicted that the CEO-TMT wage gap weakened the positive impact of CSR on innovation activities, such as R&D expenditure and patent acquisition, because a large wage gap within the TMT may be detrimental to achieving organizational objectives in organizations wherein cooperation and collaboration are important [76,77]. We found limited evidence supporting the positive synergistic effect between CSR and TMT horizontal pay equity. Finally, we predicted the potential interactive effect of CSR and CEO-employee wage gap on innovation activities. The rationale behind this is that the wider wage gap between CEOs and employees can attenuate the positive impact of CSR on innovation activities. According to the social exchange perspective, information-sharing and commitment for internal stakeholders can be weakened, resulting in a negative impact on innovation [61,66,78]. We found strong evidence of an interactive relationship between CSR and CEO-employee wage gap, implying a good match between CSR and low CEO-employee wage gap.

Based on these results, this study makes the following contribution. First, the prior study on the effects of CSR on innovation has been characterized by mixed findings. Several studies call for the needs to extend CSR-innovation link [16,17,21]. This empirical inconsistency on the effects of CRS indicate that firms differ widely in their ability to manage the impact of CSR in firms pursue innovation strategies. However, few studies have directly considered how such relationship may be changed. Thus, we explored the possibility that CEO incentives can moderate the effect of the CSR-innovation relationship. Our findings indicate that CEO compensation is one of significant variables in understanding the CSR-innovation link and a varying component in the CEO compensation system plays differential roles in affecting the relationship. Our findings are consistent with prior research in that the CEO ultimately decides whether or not to engage in CSR activities and is incentivized to actively conduct CSR [34,36,79]. The decision made by CEOs to participate in CSR is an important factor in CSR activity performance [80]. Based on our findings, the higher the CEO's performance and stock-based pay, the higher the likelihood that the CEO will make an active decision regarding long-term strategic initiatives including CSR and innovation. Therefore, this study confirmed that the compensation structure for CEOs should be designed to drive long-term and risk-taking induced management performance [67,71].

Furthermore, our findings suggested how the equity perspective and social exchange relationship from internal stakeholders or employees operate in the context of CSR-innovation relationship. Our results indicated that pay equity within and between hierar-

chies create a detrimental work environment because of the high degree of equity concern and the breach of social exchange in organizations. Specifically, the wage gap between the CEO and the TMT may be detrimental to achieving organizational activities—such as CSR and innovation—for which cooperation and collaboration are important [76,77]. Our study concluded that widening this horizontal wage gap impedes collaboration and negatively affects organizational performance [38,81]. In addition, it is necessary for corporate management to examine how employees view CSR activities, and how this perception can have a positive effect on the organization's CSR activities [82]. However, when employees perceive inequity due to differences in compensation, the cooperation, information-sharing, and commitment to the business would decrease, negatively impacting innovation activities [61]. To ensure the success of CSR initiatives, companies must, therefore, reduce the perception of unfairness in wages, build employee trust for the CEO, and develop a positive perception of organizational CSR among employees, who are the internal stakeholders.

In conclusion, this research contributes important insights to the CSR-innovation literature from the strategic human resource management perspective. Specifically, we observed that the design in executive compensation is critical that aligns the core values embedded in the CSR and innovation activities. In addition, employees' perception of equity and social exchange play a key role in affecting CSR-innovation link. This study is a first step in linking HR variables to CSR-innovation strategic management. Moreover, these findings contribute practically to our understanding of managing executive compensation. The findings can help the firms to make strategic compensation design, which can magnify the CSR-innovation relationship.

This study also has limitations that suggest the needs for further study and refinements. First, this research mainly used publicly available archival sources by estimating the economy-wide average effect of CSR on innovation and did not directly test the underlying mechanism through which employees' equity perception and social identity through compensation system can play important roles in influencing CSR on innovation. Some improvements can be made in future studies by using case analysis and perceptual employee perceptual survey to convey a more comprehensive understanding of the relationship. Secondly, future research can extend to taking a long-term oriented innovation measurement to examine its long-term effect of CSR on innovation. Additionally, a multi-level method to relate CSR initiatives to employee innovative behaviors and attitudes. Moreover, researchers need to investigate the impact of compensation practices on the process of social exchange. Finally, future avenues of research may investigate the long-term effect of CSR, elucidating the link between CSR and innovation activities, and the theoretical links between CEO compensation and corporate strategy can provide more meaningful results and implications. These are just a few examples that researchers attempt to answer for the CSR and innovation relationships.

**Author Contributions:** B.-K.C. and M.-C.C. developed the theoretical approach to CSR-innovation relationship. J.-Y.A. developed the theoretical model to the main hypothesis and the interaction effect of CEO compensation design on innovation. B.-K.C. conducted data collection and the empirical analysis. All authors jointly developed and supported the research model and relationships hypothesized. J.-Y.A. and M.-C.C. contributed to the conclusions, as well as writing, reading, and improving the final manuscript. All authors have read and agreed to the published version of the manuscript.

**Funding:** This research received no external funding.

**Institutional Review Board Statement:** Not applicable.

**Informed Consent Statement:** Not applicable.

**Data Availability Statement:** Publicly available datasets were analyzed in this study.

**Conflicts of Interest:** The authors declare no conflict of interest.

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
