# Peer review of "Corporate Social Responsibility, CEO Compensation Structure, and Corporate Innovation Activities"

_sustainability, doi:10.3390/su132313039_

Round 1

Reviewer 1 Report

This is a generally well written and organized paper on an area that is quite timely and relevant. Some thoughts on areas to be improved:

  1. Literature-Some of the areas are not directed connected to the issue of corporate innovation. The literature for various sub-sections in the literature can be slightly adjusted to ensure alignment. On a positive note the review of the literature is well organized to justify the variables explored in the study.
  2. Theoretical framework-what is discussed in this section is the conceptual model with a key discussion on CSR and innovation. Yet in the discussion, there is reference is equity and exchange theories, which is not discussed in this sub-section. Yet, these two theoretical perspectives are indeed quite relevant and would strengthen this section and that of the discussion. A suggestion is for a revision of this section and further consideration of these two theories, which are sporadically referred to in the paper.
  3. Focus on Korean Firms-This presents an interesting case, but there is no rationale to support the examination of these firms within the exploration of CSR-Innovation.
  4. Methods-much of this work is based on secondary data. It is important to state the limitations of using this in making the conclusions
  5. Variables-The definition for owner presence does not seem to align. how is status for instance translate to presence? This needs to be clearer.
  6. Data Analysis-there is the use of inferential statistics for this paper. How was normality established in this paper to justify the tests being used. This can be added in the revision of the paper.

Author Response

Response to Reviewer 1 Comments

Thank you for reviewing our manuscript entitled “Corporate social responsibility, CEO compensation structure, and Corporate innovation activities” for Sustainability. We were very pleased with the positive and constructive feedback that we received from you. Thank you for taking the time and effort necessary to provide such helpful guidance; we highly appreciate that.

Point 1: Theoretical framework-what is discussed in this section is the conceptual model with a key discussion on CSR and innovation. Yet in the discussion, there is reference is equity and exchange theories, which is not discussed in this sub-section. Yet, these two theoretical perspectives are indeed quite relevant and would strengthen this section and that of the discussion. A suggestion is for a revision of this section and further consideration of these two theories, which are sporadically referred to in the paper.

Response 1: Thank you for this valuable comment that encouraged us to elaborate more explicitly on the issue. We have clarified this issue in the section 2.2.3. and 2.2.4. and added the following sentence.

According to equity theory, individuals rethink the inputs (e.g., time, effort) to their work and the outputs (e.g., rewards) they receive from that work, and consider other tasks and inputs similar to themselves. In comparing ratios, social comparisons are made based on observable differences, even among executives within the TMT [44,39]. This comparison demonstrates that when TMT members have strong similarities in performance, employment success, and hierarchical position within the firm [42], management overestimates the value of their contributions despite differences such as the quality and quantity of their work, and tends to underestimate the value of the efforts of others [39]. High wage disparities among TMT executives are associated with emotions such as negativity, apathy, social distancing, and jealousy, and with decreased job satisfaction and a high turnover rate [39,45]. (p.5)

Wage comparison among members of an organization influences identification and motivation through social identity by evaluating the adequacy of social exchange relationships [58,59]. According to social exchange theory, the action of an employee is related to the reaction of the management through reciprocity [57]. When a CEO in the organization fails to maintain a balance by not fulfilling its obligation, breach of psychological contract occurs, which can result in negative attitudes such as job dissatisfaction, turnover intention, and negative behaviors [59]. (p.6)

Point 2. Focus on Korean Firms-This presents an interesting case, but there is no rationale to support the examination of these firms within the exploration of CSR-Innovation.

Response 2. Thank you for your suggestion. We have added more elaboration of the inclusion of the Korean sample as follows:

Moreover, the CSR-innovation link has not received much attention in the Korean context. However, anecdotal evidence suggests that there is a growing number of listed companies in Korea that utilize CSR as a driving force for innovation, thereby enhancing the company’s competitive advantage [5]. Thus, one purpose of the study is to empirically investigate the economy-wide average effect of CSR initiatives on innovation by examining all publicly traded firms in Korea. (p.1)

Point 3. Methods-much of this work is based on secondary data. It is important to state the limitations of using this in making the conclusions

Response 3. Thank you for your suggestion. We added the limitations of using a secondary dataset. In the revised manuscript, we supplemented the following.

This study also has limitations that suggest the need for further study and refinements. First, this research mainly used publicly available archival sources by estimating the economy-wide average effect of CSR on innovation and did not directly test the underlying mechanism through which employees’ equity perception and social identity through compensation system can play important roles in influencing CSR on innovation. Some improvements can be made in future studies by using case analysis and perceptual employee perceptual survey to convey a more comprehensive understanding of the relationship (p.14)

Point 4 The definition for owner presence does not seem to align. how is the status for instance translated to presence? This needs to be clearer

Response 4. Thank you very much for the helpful comment. We elaborated and rewrote the label of the variable owner-manager, indicating he/she is CEO and the largest shareholder in the company (owner).

Point 5 Data Analysis-there is the use of inferential statistics for this paper. How was normality established in this paper to justify the tests being used. This can be added in the revision of the paper

Response 5. In addition, we conducted normality tests such as Shapiro-Wilk normality tests, indicating the normal distribution of data. The following is the empirical specification model used in this study (p.8)

Reviewer 2 Report

I checked the article with the iThenticate plagiarism detection system. Plagiarism not detected. 
The topic of the article is interesting. The structure complies with the recommendation for authors.
The content is sufficiently described and contextualized with respect to previous papers on the topic. The relevance of the research is revealed and substantiated. The main purpose is presented clearly. Interesting and logical hypotheses are raised. They are based on theoretical background and stated clearly. 
However, the research methods used are not sufficiently presented. The section “Sample selection and data collection“ is written clearly. The sample (718 companies) is sufficient. 
Section „Research model“ has to be expanded. Many models can be adapted for panel data. The selection models should be described more clearly. Formed models have to be provided. 
The results are presented quite clearly. They could be compared to the results of previous research (where possible) or theory. The discussion looks logical. Conclusions are thoroughly supported by the results presented in the article. 
I agree that research results have „practical implications for establishing and implementing a suitable human resource system to improve the organizational competitiveness“.
The citation of newer scientific literature is missing. 

Author Response

Response to Reviewer 2 Comments

Thank you for reviewing our manuscript entitled “Corporate social responsibility, CEO compensation structure, and Corporate innovation activities” for Sustainability. We were very pleased with the positive and constructive feedback that we received from you. Thank you for taking the time and effort necessary to provide such helpful guidance; we highly appreciate that.

Point 1: Section „Research model“ has to be expanded. Many models can be adapted for panel data. The selection models should be described more clearly. Formed models have to be provided.

Response 1: Thank you for this valuable comment that encouraged us to elaborate more explicitly on the issue. We have revised in the following:

In this study, we conducted panel data analyses. Panel data models provide information on individual firm (or CEO) behavior, both across individual firms and over time. The data and models have both cross-sectional and time-series dimensions. The panel data we used is unbalanced when firms are not observed in all time periods. As endogeneity may arise when estimating such a panel regression model, the Hausman test was performed to detect endogeneity between these explanatory variables and individual effects. Hausman test results can identify whether the model is random-effect or fixed-effect models. If the null hypothesis cannot be rejected, there is no explanatory variable or endogeneity problem. If the null hypothesis is rejected, it should be estimated using a fixed-effect model. In addition, we conducted normality tests such as Shapiro-Wilk normality tests, indicating the normal distribution of data. The following is the empirical specification model used in this study.

ln[R&D ??+1]= β1 CSR ?? + β2 CEO performance pay ratio ??+β3 (CSR * CEO performance pay ratio)??+ β4 (Controls) ??+ ↋??                                                                                                                                                                            Point 2: The results are presented quite clearly. They could be compared to the results of previous research (where possible) or theory.

Response 2: Thank you for your suggestion. We have added the recent update on this literature and added these in the following:

The results present the more that companies engage in CSR activities, the more they conduct diverse social contribution activities utilizing the company’s core competencies [68]. This is consistent with recent findings by Zhou et al [69], suggesting CSR had a positive effect on the service and product innovation of manufacturing companies in China. Similarly, a recent study conducted using a multi-national sample suggested that the impact of CSR on corporate innovation is more pronounced in developed countries including Korea [70].

The results may imply that compensation schemes based on stock price performance are long-term rewards that may motivate managers to look beyond the short-term perspective and make long-term decisions [72-74], which can align with the value of CSR, thereby leading to innovation activities. The results confirm this argument when the patent was used as a dependent variable. This is partially consistent with the prior study, arguing that the long-term portion of CEO compensation had a positive and significant effect on corporate CSR participation [75]. However, we do not find strong evidence supporting this relationship.